# DESIGN OF THE TOPOLOGY
# FOR CONTRASTIVE VISUAL-TEXTUAL ALIGNMENT

## ABSTRACT

Pre-training weakly related image-text pairs in the contrastive style shows great power in learning semantic aligning cross-modal models. The common choice to measure the distance between the feature representations of the image-text pairs is the cosine similarity, which can be considered as the negative inner product distance of features embedded on a sphere, mathematically. However, empirically, aligning image-text pairs on the spherical topology is vulnerable to the semantic ambiguity phenomenon resulting from the noise in the pre-training datasets. Specifically, under the noisy training data, instead of the optimal alignment-uniformity solution, the system would achieve an equilibrium (a gap between distances of positive and negative pairs), when the gradients for attraction and repulsion are neutralized. Although intuitively, the model should always find this equilibrium given a sufficiently long training scheme. However, since the contrastive loss is implemented using softmax and cross-entropy loss, which makes the required numerical values for equilibrium much larger, where the equilibrium might be out of the distance range (*e.g.* [-1, 1] for the cosine similarity). In the practice of former studies, this problem is partly tackled by introducing a learnable softmax temperature parameter, in other words, by explicitly scaling the range of the distance function. In this work, we alternatively design the topology of embedding space and its endowed distance function. Motivated by studies that make use of Riemannian geometry for visual tasks, we propose a rather simple solution to address the aforementioned equilibrium problem. That is, we map the feature representations onto the oblique manifold endowed with the negative inner product as the distance function. Furthermore, we propose a multi-token implementation of the oblique manifold. With this configuration, we outperform the officially released CLIP-ViT/L-14 model using a ViT/B-16 visual backbone on the zero-shot image classification and image-to-text retrieval tasks.

## 1 INTRODUCTION

Learning visual and textual feature representations that are semantically aligned in their embedding space is an ordinary problem in the vision-language cross-modal tasks (Frome et al., 2013; Karpathy & Fei-Fei, 2015; Romera-Paredes & Torr, 2015; Wang et al., 2016; Faghri et al., 2017; Xian et al., 2016). In early works that employ feature representations from deep neural networks, *e.g.* Frome et al. (2013), the alignment is often achieved by a fundamental metric learning approach with the hinge rank loss. That is, the similarity between a visual feature vector $u$ and a textual feature vector $v$ is calculated as $u^T W v$, where $W$ are the learnable weight parameters. Thanks to the revolutionary advances in computational power, we can now achieve this in a more effective and practical approach termed contrastive learning, where we align quantities of positive samples and push their negative samples away simultaneously in a large mini-batch (Radford et al., 2021; Singh et al., 2022; Jia et al., 2021; Pham et al., 2021; Yuan et al., 2021). The standard choice of the distance measure between an image-text pair for the contrastive learning algorithm is the `Cosine Similarity` (in both uni-modal Chen et al. (2020a); Caron et al. (2020); Chen et al. (2020b) and cross-modal Radford et al. (2021); Jia et al. (2021); Singh et al. (2022) scenarios). Mathematically, the `Cosine Similarity` computes the negative inner product value between feature representation vectors mapped onto the unit sphere embedding space. The spherical embedding space is advantageous in aligning visual and textual feature representations in the following two aspects. First, calculat-

ing the inner product consumes low computational resources during both forward and backward propagation. Second, we have a proper definition of uniformity on the sphere. The uniformity is a demanded property in optimizing the contrastive loss, by which the feature representations would preserve maximal information of the data.

However, since the data for large-scale contrastive alignment are internet-collected noisy image-text pairs, we often find pairs of semantically related images and texts labeled as "negative" and verse visa, which we term "semantic ambiguity". Because of the ambiguity, it is impossible to achieve the perfect alignment and uniformity conditions of sample embeddings for the system. More specifically, during the training, the false negative samples are pushed away from each other (repulsion), while the false positive samples are pulled together (attraction). As a consequence, the system will gradually find an equilibrium when the noisy samples' gradients for attraction and repulsion are neutralized. In other words, we say the training progress is *converged* under the given hyper-parameters. To be more concrete, owing to the fact that the gradient is eventually back-propagated from the difference between the positive and negative distances. Given sufficient model capacity, the numerical values between the distances of positive and negative pairs of samples will be optimized to fit the noisy level of the dataset. For instance, if there is a reasonable amount of false negative samples, then the model would learn a smaller positive distance for not being punished too hard, when encountering false negative samples in another mini-batch. Furthermore, the triangular inequality (or a "relaxed" version) of the distance within a group of semantically similar samples will pull the positive pairs of samples away from each other (See Section 3.2). Finally, the model reaches the equilibrium of compromised positive and negative distances, which minimizes the contrastive loss regardless of the semantic ambiguity. Therefore, we say the equilibrium is essentially a property of the dataset, referring to its noise level, under a certain embedding space and its distance function.

Here, the problem is that, since we implement the contrastive loss with the combination of softmax and cross-entropy, which makes the required numerical values for equilibrium exponentially larger. The numerical values of distances required by softmax at equilibrium might be out of the distance range (*e.g.* [-1, 1] for the cosine similarity). Therefore, in the practice of former studies (Wu et al., 2018; Radford et al., 2021), researchers have to scale the range with a learnable softmax temperature parameter. Although the learnable temperature somehow fixes the range problem, it still has two drawbacks that need to be addressed. Firstly, the learnable temperature delays the learning progress. Empirically, we observe that the model trends to acquire a proper scaling for the distance range earlier than achieving a good alignment. Secondly, a large temperature is numerically unstable for the back-propagation, especially for the low-bit precision computation. In this work, we alternatively design the topology for embedding vectors and its endowed distance function. Motivated by the utilization of Riemannian geometry for visual tasks, we propose a relatively simple solution to address the aforementioned out-of-range equilibrium problem. Our contributions can be summarized as follows:

1. We reveal that the learnable softmax temperature is essentially a scaling factor for the distance range, which indicates the noise level of the dataset in the contrastive visual-textual alignment. We also observe that the model learns a suitable temperature before representations, which degrades the performance.

2. We tackle the out-of-range equilibrium problem resulting from the softmax cross-entropy loss, by designing the topology of embedding space. That is, we employ the oblique manifold endowed with the negative inner product as distance functions.

3. We demonstrate that the proposed approach can be non-painfully implemented by changing only two lines of the training code, whilst improving the baseline performance in the zero-shot image-text retrieval tasks. In the larger scale experiment, we have learned a ViT-B/16 model that outperforms the officially released ViT-L/14 model.

## 2  PRELIMINARY

**Notations:** We start with notation and review mathematical expressions of the basic building blocks used in our analysis. In this work, we denote scalars by italic letters, *e.g.,* $n, m, B, D \in \mathbb{R}$, and denote vectors and higher-order tensors by boldface letters, *e.g.,* $\mathbf{x} = [x_0, x_1, \ldots, x_{n-1}]^\top \in \mathbb{R}^n$ and $\mathbf{Y} \in \mathbb{R}^{N \times D}$. We denote sets by calligraphic letters, *e.g.,* $\mathcal{U} = \{\boldsymbol{U}_1, \boldsymbol{U}_2, \ldots\}$. We also employ italic letters to define functions, with subscripts denote their parameters, *e.g.,* $f_\theta(\cdot)$. The operation $\| \cdot \|_p$

denotes the $\ell_p$ norm of a vector and $|\cdot|$ denotes the absolute value of a scalar. For any integer $K$, we use $[K]$ to denote the set of integers from 1 to $K$.

**Visual-Textual Pre-trained Model:** Given a set of semantically related image-text pairs $\mathcal{S} = \{(\boldsymbol{U}_1, \boldsymbol{V}_1), (\boldsymbol{U}_2, \boldsymbol{V}_2), \ldots, (\boldsymbol{U}_K, \boldsymbol{V}_K)\}$, where $\boldsymbol{U}$ is an image of size $H \times W \times C$, $\boldsymbol{V}$ is a tokenized text of length $L$. The goal is to simultaneously learn a pair of encoders $f_\theta : \boldsymbol{U} \to \boldsymbol{u}, g_\phi : \boldsymbol{V} \to \boldsymbol{v}$ to map the image and text into an embedding space, $\boldsymbol{u}, \boldsymbol{v}$ are called embedding vectors of samples. A well-optimized visual-textual pre-trained model aligns the embedding vectors across the visual and textual models. That is, the embedding vectors extracted semantically related image-text pairs earn higher similarity scores than the non-related ones. To generalize the problem, we view the embedding vectors as points on specified typologies, and the similarity score between embedding vectors is an endowed distance function that evaluates the distance between the points. For instance, the commonly employed `cosine similarity` calculates the inner product of the normalized embedding vectors on the unit sphere as the (negative) distance between the sample pairs. To this end, we further consider the encoders as compositions of functions that i) map the inputs into the euclidean space and ii) map the input vectors in euclidean space on specified typologies. We denote this two-step mapping as: $f_\theta = \bar{f}_\theta \cdot f$, where $\bar{f}_\theta : \boldsymbol{U} \to \bar{\boldsymbol{u}}$ is the encoder with learnable parameters, $\bar{\boldsymbol{u}} \in \mathbb{R}^d$ is the output in the $d$-dimensional euclidean space. $f : \bar{\boldsymbol{u}} \to \boldsymbol{u}, \boldsymbol{u} \in \mathcal{M}$ is an specified operator (without learnable parameters) maps the representations onto a topology $\mathcal{M}$, which is usually considered as a manifold embedded in the $d$-dimensional euclidean space.

**Contrastive Learning:** Following the definition in Oord et al. (2018); Wang & Isola (2020); Chen et al. (2021a); Radford et al. (2021), we formulate the contrastive loss as

$$\mathcal{L}_c(f_\theta, g_\phi; \tau, \mathcal{S}) := \mathop{\mathbb{E}}_{\substack{\boldsymbol{U}, \boldsymbol{V} \sim \mathcal{S} \\ \boldsymbol{U}_i^- \neq \boldsymbol{U} \\ \boldsymbol{V}_j^- \neq \boldsymbol{V}}} \left[ -\log \frac{e^{-\tau d(f_\theta(\boldsymbol{U}), g_\phi(\boldsymbol{V}))}}{\sum_{j \in [M]} e^{-\tau d(f_\theta(\boldsymbol{U}), g_\phi(\boldsymbol{V}_j^-))} + \sum_{i \in [M]} e^{-\tau d(f_\theta(\boldsymbol{U}_i^-), g_\phi(\boldsymbol{V}))}} \right], \tag{1}$$

where $\tau$ is the temperature term, we write it as a multiplier for simplicity. $d(\cdot, \cdot)$ is the distance function between two points. $M \in \mathbb{Z}^+$ is a fixed number of negative samples. Briefly, optimizing this loss term minimizes the distance between positive image-text pairs and maximizes the distance between negative image-text pairs. It is worth mentioning that, in recent studies Radford et al. (2021); Chen et al. (2021b), the contrastive loss is usually implemented as the cross-entropy between one-hot labels and the class probability obtained by `softmax` within a mini-batch $\mathcal{S}_M$. We also employ this implementation in this work, which can be formulated as

$$\mathcal{L}_c(f_\theta, g_\phi; \tau, \mathcal{S}) = \mathop{\mathbb{E}}_{\substack{\boldsymbol{U}, \boldsymbol{V} \sim \mathcal{S}_M \\ i \in [M]}} \left[ H\Big(\boldsymbol{q}_i | \sigma(\mathcal{U}_i)\Big) + H\Big(\boldsymbol{q}_i | \sigma(\mathcal{V}_i)\Big) \right], \tag{2}$$

where $H(\cdot|\cdot)$ is the cross-entropy loss, $\mathcal{U}_i = \{-\tau d(f_\theta(\boldsymbol{U}_i), g_\phi(\boldsymbol{V}_j))\}_{j \in [M]}$, $\mathcal{V}_i = \{-\tau d(f_\theta(\boldsymbol{U}_j), g_\phi(\boldsymbol{V}_i))\}_{j \in [M]}$ are the (negative) distance between an image/text to all the texts/images in the mini-batch. $\sigma$ is the `softmax` function, $\boldsymbol{q}_i$ is the one-hot label vectors of $i$.

**Oblique Manifold:** Briefly, the oblique manifold $\mathrm{Ob}(n, n)$ is the set of matrices of size $n \times m$ with unit-norm columns. We follow the definition in Absil et al. (2009), that is,

$$\mathrm{Ob}(n, m) := \{\boldsymbol{X} \in \mathbb{R}^{n \times m} : \mathrm{diag}(\boldsymbol{X}^T \boldsymbol{X}) = \boldsymbol{I}_n\}, \tag{3}$$

where the $\mathrm{diag}(\cdot)$ is the diagonal entries of the matrix, $\boldsymbol{I}_n$ is the identity matrix of size $n$. The geometry of an oblique manifold is exactly the same as that of the product manifold of spheres $\underbrace{\mathbb{S}^{n-1} \times \cdots \times \mathbb{S}^{n-1}}_{m \text{ copies}}$, where $\mathbb{S}^{n-1}$ is the sphere manifold embedded in $\mathbb{R}^n$.

## 3 METHOD

In this section, we introduce how we implement the proposed approach. That is, employing the *oblique manifold* as the embedding space, with the (negative) *inner product* as its distance function.

```
1   # image_encoder - vision transformer
2   # text_encoder  - text transformer
3   # U             - mini-batch of aligned images, [n, h, w, c]
4   # V             - mini-batch of aligned texts, [n, l]
5   # Ob_m          - m of the oblique manifold
6   # Ob_n          - n of the oblique manifold
7   # t             - learned temperature parameter
8
9   # extract features of each modality
10  u_bar = image_encoder(U) #[n, d]
11  v_bar = text_encoder(V)  #[n, d]
12
13  ''' Lines 15-16 are the only codes that need to change compared to a standard CLIP algorithm. '''
14  # map features onto Ob(n,m)
15  u = u_bar.reshape('n d -> n Ob_m Ob_n').l2_normalize(axis=-1).reshape('-> n d')
16  v = v_bar.reshape('n d -> n Ob_m Ob_n').l2_normalize(axis=-1).reshape('-> n d')
17
18  # scaled pairwise inner product distance, [n, n]
19  neg_distances = u @ v.t() * t.exp()
20
21  # symmetric loss function
22  labels = arange(n) # 0, 1, ..., n
23  loss = (CE_loss(neg_distances, labels, axis=0) + CE_loss(neg_distances, labels, axis=1)) / 2
```

Figure 1: Python-like pseudo-code for the core of the proposed approach.

## 3.1 PROPOSED APPROACH

**Basic Implementation:** To map the feature vectors onto the oblique manifold $\mathrm{Ob}(n, m)$, we first reshape the feature vectors $\bar{\boldsymbol{u}}, \bar{\boldsymbol{v}}$ of size $d$ to a matrix of shape $m \times n$, then we $\ell_2-$normalize the columns to obtain $\boldsymbol{u}, \boldsymbol{v}$. Finally, the distance is defined as the negative value of the trace of the matrix product, *i.e.* $d(\boldsymbol{u}, \boldsymbol{v}) = -\mathrm{tr}(\boldsymbol{u}^T \boldsymbol{v})$. We provide the python-style pseudo-code of the proposed approach in Figure 1. It is worth mentioning that, in the implementation, we vectorize the embedding vectors back to their original size $d$, then apply the vector inner product to compute the distance. Moreover, we employ the term "neg_distances" to avoid reduplicated calculation of the negative operation.

**Multi-Token Implementation:** We provide another implementation of the proposed topology and distance functions by utilizing the [CLS] token (the token used to reperent global information in ViT, see the implementation details in Dosovitskiy et al. (2020)) of the transformer architecture. Briefly, we attach multiple randomly initialized [CLS] tokens in the first embedding layers. At the output layer, we consider each token embedding as a sub-sphere of the oblique manifold. We then compute the inner product of the vectorized visual and textual embeddings. Unlike the standard implementation, the multi-token implementation requires more computational resources in the backbone. On the contrary, the sub-sphere could benefit from the global attention operation and provide more representative feature embeddings.

## 3.2 RETHINKING THE PROPERTIES OF TOPOLOGY

In this section, we discuss the motivation for our proposed approach in detail. Besides the former discussed distance range, we further discuss the properties of the proposed approach by comparing it with three reference configurations of different topologies and distance functions. Specifically, we consider i) the sphere $\mathbb{S}^{d-1}$ endowed with the inner product as distance, ii) the euclidean space $\mathbb{R}^d$ endowed with $\ell_2$ distance; iii) the oblique manifold $\mathrm{Ob}(d/m, m)$ endowed with the minimizing geodesic as distance, which is denoted as $\mathrm{Geo}(\boldsymbol{u}, \boldsymbol{v}) = \mathrm{tr}^{\frac{1}{2}}(\arccos^2(\boldsymbol{u}^T \boldsymbol{v}))$. The overview of the comparison is summarized in Table 1. The comparison summarizes the three important properties of the embedding topology in addition to the distance range. Keeping these properties favored for contrastive learning is important in the design of embedding topology.

| Topology | Sphere $\mathbb{S}^{d-1}$ | Euclidean $\mathbb{R}^d$ | Oblique$(d/m, m)$ | Oblique$(d/m, m)$ |
|---|---|---|---|---|
| Distance | $-\boldsymbol{u}^T\boldsymbol{v}$ | $\|\boldsymbol{u} - \boldsymbol{v}\|_2$ | $\mathrm{Geo}(\boldsymbol{u}, \boldsymbol{v})$ | $-\mathrm{tr}(\boldsymbol{u}^T\boldsymbol{v})$ |
| Memory Resource | $O(b^2)$ | $O(b^2 d)$ | $O(b^2 m)$ | $O(b^2)$ |
| Uniformity | surface measure | undefined | surface measure | surface measure |
| Inequality | relaxed | restricted | restricted | relaxed |
| Distance Range | $[-1, 1]$ | $[0, +\infty)$ | $[0, m\pi]$ | $[-m, m]$ |

Table 1: Summary of different topologies endowed with different distances. The total dimension of the embedding vector is denoted as $d$. The mini-batch size is denoted as $b$. Green box stands for the properties that are *favored* for contrastive learning. Red box stands for the properties that are *unfavored* for contrastive learning. Best view in color.

**i. Low computational resource:** In the contrastive learning algorithm, the logits (or distance/similarity) matrix often costs the most computational resource in large-scale training. For instance, given a mini-batch of sample pairs of size $b$ with $d-$dimensional output, the computation of the inner product achieves a complexity of $O(b^2 d)$ and storage usage of $O(b^2)$. However, since the back-propagation of the $\ell_2-$norm requires intermediate results that cannot be "inplace" calculated, the $\ell_2-$norm (or any $\ell_p-$norm based distance) in euclidean space requires a storage usage of $O(b^2 d)$. On the contrary, the geodesic distance cache $m$ curve lengths and hence requires $O(b^2 m)$ storage usage, while the proposed approach only needs one matrix multiplication after re-vectorization.

**ii. Proper definition of uniformity:** As explained by Wang & Isola (2020), contrastive loss is a combination of two objects a) alignment of features from positive sample pairs and b) the distribution of the features encouraged to match. Naturally, with the loss form defined in Equation (2), the distribution object will result in a uniform distribution on the sphere. Although it is not essential for the distribution of samples to be uniform as discovered by Chen et al. (2021a), it is necessary to define a proper prior distribution for samples to match via optimal transport algorithms (*e.g.* sliced Wasserstein distance), which is undoubtedly a computational burden. Both the sphere and oblique manifold have a proper uniform distribution defined as the surface area measure, while the unbounded euclidean space does not. In practice, the $\ell_2-$norm defined distance between the samples grows larger along training and eventually overflows.

**iii. "Relaxed" triangular inequality:** Assume that we have a "well-optimized" model $f_\theta^*, g_\phi^*$, where for a positive pair $(\boldsymbol{U}, \boldsymbol{V})^+$, their distance $\epsilon^+ := d(\boldsymbol{u}^*, \boldsymbol{v}^*) = d(f_\theta^*(\boldsymbol{U}), g_\phi^*(\boldsymbol{V}))$ is reasonable small, and the distance $\epsilon^-$ for negative pairs $(\boldsymbol{U}, \boldsymbol{V})^-$ is reasonable large. Let us consider the following scenario, in $\mathcal{S}_\pm = \{(\boldsymbol{U}_1, \boldsymbol{V}_1), (\boldsymbol{U}_2, \boldsymbol{V}_2)\}$, the pair $(\boldsymbol{U}_1, \boldsymbol{V}_2)^-$ is also semantically correlated despite of being recognized as a negative sample. For the "well-optimized" model, it will predict a distance of $\epsilon^+$ for this pair instead of $\epsilon^-$. If the distance function $d$ is a *metric*, then according to the triangle inequality axiom of metric, we have the following inequality,

$$\epsilon^- = d(\boldsymbol{u}_2^*, \boldsymbol{v}_1^*) \leq d(\boldsymbol{u}_1^*, \boldsymbol{v}_2^*) + d(\boldsymbol{u}_2^*, \boldsymbol{v}_2^*) + d(\boldsymbol{u}_1^*, \boldsymbol{v}_1^*) = 3\epsilon^+ \tag{4}$$

We can see that the negative distance is bounded by three times the positive distance, despite the fact that the positive distance should be as small as possible. Given the fact that, in a mini-batch of sufficiently large size, it is very likely to have negative pairs of image and text that match each other equally well as the positive ones. Therefore, we need to "relax" the triangular inequality to alleviate the effects of ambiguity of the positive/negative pairs. In practice, we have the following observations: i) The positive pairs of samples usually have a much larger distance than that in the perfect alignment condition, ii) $d(\boldsymbol{u}_1^*, \boldsymbol{u}_2^*)$ may become exceptionally small since none of the loss terms regularize it, resulting in a further tightened bound of the negative distance; iii) Although the inner product distance is not a *metric*, it still obeys an "relaxed" triangular inequality because we can yield a metric on the sphere by `ArcCos` (the geodesic), see Schubert (2021) for more information. In Table 1, we characterize both the minimizing geodesic and the $\ell_2-$norm distances are metrics and hence obey the restricted triangular inequality.

## 4 EXPERIMENTAL ANALYSIS

### 4.1 EXPERIMENTAL SETTINGS

**Datasets:** For the experimental analysis in Section 4.2, we employ the 15M subset (Cui et al., 2022) of the YFCC100M dataset (Thomee et al., 2016) as the training dataset, which contains roughly 15.3 million internet collected weakly related image-text pairs. We evaluate the proposed methods with two types of vision tasks: i) `Zero-Shot` image-to-text and text-to-image retrieval on Flickr30k (Plummer et al., 2015) and ii) `Zero-Shot` and `Linear Probe` object classification on ImageNet-1K (Russakovsky et al., 2015). Furthermore, we employ the RedCaps (Desai et al., 2021) dataset as the out-domain data for visualizing the distributions of sample distances. For the experimental analysis in Section 4.2, we collect data from publicly available datasets (Schuhmann et al. (2021); Changpinyo et al. (2021); Sharma et al. (2018); Chen et al. (2015); Krishna et al. (2017); Plummer et al. (2015); Chen et al. (2015); Russakovsky et al. (2015); Desai et al. (2021); Kuznetsova et al. (2020); Li et al. (2017), etc.). We also have clawed weakly related image-text pairs from the web, resulting a total of 420 million individual images and roughly 500 million image-text pairs. This dataset is comparable to the one employed in the official CLIP paper (Radford et al., 2021) and another open source re-implementation (Ilharco et al., 2021). To further remove the bias caused by datasets, we also re-implement the naive clip algorithm for reference.

**Models:** Due to the limited computational resources, we adopt a moderate scaling of the models. Specifically, For the ablation experiments, we employ the original ViT-S/16 architecture for our image encoders (Dosovitskiy et al., 2020), with an input image resolution of 224, resulting in 196 image tokens. For large-scale training, we employ the ViT-B/16 as our image encoders. For our text encoders, we employ Ernie-2.0-en-base (Sun et al., 2020), which is literally a Bert model (Devlin et al., 2018) of 12 layers and 512 hidden neuron sizes with a customized vocabulary of 30,522 tokens, and the maximum context length is set to be 77. We project the feature representation ([CLS] token) from the top layer of transformers to a (sum of) 512-dimensional embedding space. All the parameters except the temperature are optimized from random initialization. The default initialization of the project matrix employs the Gaussian initializer of zero mean, and standard deviation equal reversed square root of the input size (*a.k.a.* Kaiming initialization). For the temperature, we initialize it with $e^t$ for $t = e^0, e^1, e^2$. Hyper-parameters employed for training are provided in the appendix. The details of the hyperparameters are provided in Table 7 in the appendix.

**Evaluation:** For zero-shot retrieval on Flickr30K and Coco, we employ the logits (distance) computed by the distance function and report the image-text pairs with the top-$k$ shortest distance as the retrieval results. For zero-shot classification on ImageNet. We employ multiple prompt templates described in Radford et al. (2021), while we first compute the distances between image and text embeddings, then average the distances. For linear probe classification on ImageNet, we remove the learned projection head (no topological structure is preserved), then attach a random initialized linear projector to map the feature representation to the 1,000 class logits.

### 4.2 EXPERIMENTS RESULTS

**Comparison between different configurations:** In Table 2, we report the performance of the configurations described in Table 1. We employ the oblique manifold structure of $n = 64, m = 8$ for our proposed approach, denoted as Ob(64,8) in the table. The proposed approach provides +4.0%/1.44% improvement over the baseline for the retrieval tasks. It is worth noting that the geodesic distance version also improves the baseline reasonably, despite the fact that it obeys the restricted triangular inequality. This suggests that a more elaborate topology structure helps the alignment of the features from different modalities. The Euclidean configuration performs the worst, which could be the drawback of having no properly defined uniformity. For the zero-shot and linear probe classification accuracy on ImageNet, the oblique configurations provide better performance when the temperature initialization is low. We suggest the reason is that the equilibrium temperature for the oblique is lower than that for the sphere; hence the model could focus more on learning the representations of the visual modality.

**Effects of temperature initialization:** In Table 2, we also demonstrate the results using different temperature initialization. Although it can be seen that the final performance is not largely impacted by the initialization, a large or small temperature at the start of the training still cause difficulties

| Topology | Distance | Zero-Shot I2T R@1 | Zero-Shot T2I R@1 | Zero-Shot Cls. Acc. | Linear Probe Cls. Acc. |
|---|---|---|---|---|---|
| Temperature, init=$e^1$, gradient=True | | | | | |
| Sphere | $-\boldsymbol{u}^T\boldsymbol{v}$ | 48.3 | 31.45 | 30.62 | 60.38 |
| Euc | $\|\boldsymbol{u}-\boldsymbol{v}\|_2$ | 47.9 | 32.29 | 30.36 | 59.90 |
| Ob(64, 8) | $\text{Geo}(\boldsymbol{u},\boldsymbol{v})$ | 50.7 | 32.35 | **30.79** | 60.43 |
| Ob(64, 8) | $-\text{tr}(\boldsymbol{u}^T\boldsymbol{v})$ | **52.3** | 32.89 | 30.70 | 60.32 |
| Temperature, init=$e^2$, gradient=True | | | | | |
| Sphere | $-\boldsymbol{u}^T\boldsymbol{v}$ | 46.8 | 31.13 | 29.60 | 59.82 |
| Euc | $\|\boldsymbol{u}-\boldsymbol{v}\|_2$ | 48.5 | 32.69 | 30.68 | 59.74 |
| Ob(64, 8) | $\text{Geo}(\boldsymbol{u},\boldsymbol{v})$ | 50.7 | 31.59 | 30.34 | 59.60 |
| Ob(64, 8) | $-\text{tr}(\boldsymbol{u}^T\boldsymbol{v})$ | 50.3 | **33.37** | 30.23 | 59.76 |
| Temperature, init=$e^0$, gradient=True | | | | | |
| Sphere | $-\boldsymbol{u}^T\boldsymbol{v}$ | 49.0 | 30.33 | 28.59 | 59.56 |
| Euc | $\|\boldsymbol{u}-\boldsymbol{v}\|_2$ | 47.4 | 30.71 | 29.85 | 60.09 |
| Ob(64, 8) | $\text{Geo}(\boldsymbol{u},\boldsymbol{v})$ | 49.9 | 32.49 | 30.21 | 60.61 |
| Ob(64, 8) | $-\text{tr}(\boldsymbol{u}^T\boldsymbol{v})$ | 50.9 | 32.71 | 30.50 | **60.66** |
| Temperature, init=$e^0$, gradient=False | | | | | |
| Sphere | $-\boldsymbol{u}^T\boldsymbol{v}$ | 5.1 | 3.461 | 4.04 | 45.37 |
| Euc | $\|\boldsymbol{u}-\boldsymbol{v}\|_2$ | 47.6 | 30.43 | 29.51 | 59.20 |
| Ob(64, 8) | $\text{Geo}(\boldsymbol{u},\boldsymbol{v})$ | 4.1 | 2.921 | 3.10 | 21.67 |
| Ob(64, 8) | $-\text{tr}(\boldsymbol{u}^T\boldsymbol{v})$ | 30.3 | 18.48 | 21.20 | 57.93 |

Table 2: The retrieval and classification performance of different configurations under different temperature initialization conditions. "gradient={True/False}" donates if the temperature is learnable.

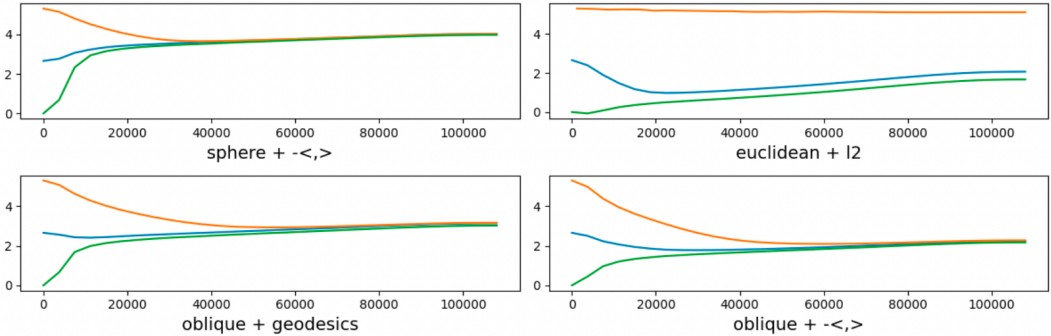

Figure 2: The learning curve of the temperature. -¡,¿, l2 and geodesics denote the negative inner product, $\ell_2$, and minimizing geodesic distance, respectively. The orange, blue and green curves denote the initialization of $e^2$, $e^1$, and $e^0$, respectively.

in optimizing. It is also interesting to see that when the temperature is fixed at 1.0, the sphere one (narrow distance range) and the geodesic one (restrict triangular inequality) fail to learn meaningful feature embeddings for contrastive alignment. The performance of the proposed approach is also largely reduced. However, since the euclidean topology does not have an upper bound of the distance, the optimizer can still reach the equilibrium. We further draw the trend of the temperature during the training progress in Figure 2. From the figures, we can confirm that: i) Given a bounded distance range, the temperature is an inherent property of the datasets, depicting the noise level of the datasets; ii) The temperature will first converge to an equilibrium regardless of initialization, then raise gradually along the optimization progress; iii) It takes longer training iterations for the temperature to converge on the oblique embedding space, while the final temperature is smaller than that in the sphere embedding space. The results suggest that border distance range and topology structure help the model focus more on aligning images and texts rather than finding the equilibrium.

Retrieval performance on Flickr and Coco of our implemented TCL model and the variant using our proposed method. The numbers in brackets are the performance obtained using the contrastive alignment head.

**Ablation on oblique structures and multi-token implementation:** In Table 3, we modify the structure of the oblique manifold under fixed total dimensions. It can be seen that a higher $m$

| Topology | Distance | Zero-Shot I2T R@1 | Zero-Shot T2I R@1 | Zero-Shot Cls. Acc. | Linear Probe Cls. Acc. |
|---|---|---|---|---|---|
| | | Temperature, init=$e^1$, gradient=True | | | |
| Sphere(512) | $-\boldsymbol{u}^T\boldsymbol{v}$ | 48.3 | 31.45 | 30.62 | 60.38 |
| Ob(256, 2) | $-\mathrm{tr}(\boldsymbol{u}^T\boldsymbol{v})$ | 48.0 | 32.25 | 30.33 | 60.52 |
| Ob(64, 8) | $-\mathrm{tr}(\boldsymbol{u}^T\boldsymbol{v})$ | **52.3** | 32.89 | 30.70 | 60.32 |
| Ob(16, 32) | $-\mathrm{tr}(\boldsymbol{u}^T\boldsymbol{v})$ | 50.4 | **33.01** | **30.93** | **60.79** |
| Ob(4, 128) | $-\mathrm{tr}(\boldsymbol{u}^T\boldsymbol{v})$ | 48.2 | 32.91 | 30.57 | 59.99 |
| Multi(256, 2) | $-\mathrm{tr}(\boldsymbol{u}^T\boldsymbol{v})$ | 49.2 | 32.29 | 30.04 | 61.59 |
| Multi(64, 8) | $-\mathrm{tr}(\boldsymbol{u}^T\boldsymbol{v})$ | 54.0 | **34.27** | **31.93** | 62.41 |
| Multi(16, 32) | $-\mathrm{tr}(\boldsymbol{u}^T\boldsymbol{v})$ | **54.0** | 33.43 | 30.88 | **63.71** |

Table 3: The retrieval and classification performance of the proposed approach using different oblique manifold structures and the multi-token implementation. "gradient={True/False}" donates if the temperature is learnable.

| Method baseline[impl.] | ImageNet Zero-shot Acc@1 | Flickr Zero-shot | | | Coco Zero-shot | | |
|---|---|---|---|---|---|---|---|
| | | I2T R@1 | T2I R@1 | Recall mean | I2T R@1 | T2I R@1 | Recall mean |
| *ViT-B/16-224 as visual bone.* | | | | | | | |
| CLIP[openAI[†]] | 68.7 | 81.9 | 62.1 | 86.1 | 55.4 | 38.4 | 66.3 |
| CLIP[openCLIP[‡]] | 67.0 | 83.2 | 65.5 | 87.6 | 52.4 | 38.4 | 62.4 |
| CLIP[our-impl.] | 69.5 | 84.2 | 61.7 | 86.4 | **64.1** | **43.9** | **72.4** |
| CLIP[Multi(16,32)] | **76.4** | **85.2** | **66.3** | **88.3** | 63.8 | 42.9 | **72.4** |
| *ViT-L/14-224 as visual bone for reference.* | | | | | | | |
| CLIP[openAI[†]] | 75.5 | 85.0 | 65.2 | 87.7 | 56.3 | 36.5 | 65.2 |
| CLIP[openCLIP[‡]] | 72.7 | 87.6 | 70.3 | 90.1 | 59.7 | 43.0 | 70.0 |

Table 4: Comparsion of large scale contrastive visual-textual pre-train model on benchmark datasets. [†] and [‡] denote the implementation from Radford et al. (2021) and Ilharco et al. (2021), respectively.

value (*i.e.* the number of product sub-spheres) is more likely to obtain a better zero-shot text-to-image retrieval recall. However, an over-complicated structure such as Ob(4,128) could ruin the performance. We conjecture that, since the sphere has one redundant dimension, the larger number of product sub-spheres reduces the representation capacity of the topology. And because we employ a textual encoder that is gently larger than the visual one, therefore the moderate reduction of the capacity helps overcome the overfitting on the textual side. We also provide the results using the multi-tokens implementation discussed in Section 3, denoted as Multi($\cdot,\cdot$). For the linear probe, we concatenate all the representations together before projecting them to 1,000 class logit. This implementation outperforms other configurations by a large margin at the cost of computational resources. Training the largest Multi(16, 32) models takes approximately 14% more time.

### 4.3 RESULTS USING LARGE-SCALE DATASET

In Table 4, we compare the performance of the proposed method using a larger scale configuration. It can be seen that using the ViT-B/16 as the visual bone, our implementation of the naive model performance is similar to the publicly released ones. On the other hand, the model equipped with the multi-token implementation of (16,32) significantly outperforms the other ViT-B/16 models, with only less than 8% more computational costs.

## 5 RELATED WORKS

**Momentum distillation:** In recent works such as Cheng et al. (2021); Li et al. (2021a), the momentum (self-)distillation is introduced to mitigate the semantic noise in the sample pairs. That is, a momentum version of the model is updated by the moving average of the model's historical parameters. Then, the cross entropy between the softmax logits computed by the model and its momentum version is used as an additional loss for supervision. The authors claim that the pseudo-targets of the momentum (self-)distillation will not penalize the model for matching negative samples that are reasonably similar. Here, we consider that the pseudo-targets do "relax" the triangular inequality

restriction implicitly by letting the distance of alignment be reasonably large. Hence, it could be much easier for the optimizer to find the equilibrium discussed in Section 3.2.

**Other implementation of non-metric distance:** In Yao et al. (2021), the authors proposed a so-called fine-grained contrastive learning scheme that matches all the visual and textual tokens using a maximum-average operator. Concretely, for each visual token, it finds the textual token with maximum similarity, then takes the average over the visual tokens as the similarity of the image to a text and vice versa. Using our framework, this work can be explained as embedding samples onto the product manifold $\mathbb{S}^{d-1} \times \cdots \times \mathbb{S}^{d-1}$ endowed with the maximum-average distance, which is a non-metric distance. At the same time, the authors employ the sub-manifold $\mathbb{S}^{d-1}$ to represent local information.

**The effects of softmax temperature:** In Wang & Liu (2021), the authors draw the uniformity of the embedding distribution and the tolerance to semantically similar samples of learned models under different temperatures. From the observations, the authors claim that "a good choice of temperature can compromise these two properties properly to both learn separable features and tolerant to semantically similar samples, improving the feature qualities and the downstream performances". Unlike our work, this work is done under uni-modal contrastive learning, where the semantic correlation of the negative samples is not a property of the datasets but rather a drawback of the larger mini-batch size.

**Uni-modal side tasks:** In works such as Mu et al. (2021); Li et al. (2021b); Yang et al. (2022), authors combine cross-modal contrastive loss with other uni-modal tasks, for instance, visual/textual self-supervised contrastive learning, masked image/language modeling. These combined methods demonstrate superior performance in downstream tasks such as zero-shot classification empirically. Although these works do not overlap with this one, we find that the uni-modal tasks provide reasonable uniformity within the visual/textual feature embedding, contrary to the cross-modal contrastive shown in Section 3.2. Therefore, the model could obtain a more "numerically relaxed" triangular inequality when dealing with noisy pairs of samples.

**Other works that employ oblique manifold:** It is notable that learning representations embedded on the oblique manifold for computer vision tasks have been explored by former studies. For instance, in Qi et al. (2021), the authors employ the oblique topology for few-shot learning. However, different from these works, our paper mainly tackles the noisy database problem in the contrastive image-text alignment task. We employ the oblique topology with a non-metric distance function to tackle the out-of-range equilibrium problem.

**Hyperbolic embedding space:** The hyperbolic topology is another popular choice for hierarchical representation embedding space, in both NLP (Nickel & Kiela, 2017), CV (Zhang et al., 2022; Liu et al., 2020; Khrulkov et al., 2020), and cross-modal (Guo et al., 2021) tasks. However, the hyperbolic topology has unfavored properties similar to Euclidean space. That is, the resource required for computing distance is high, difficult to define/implement uniformity in terms of numerical stability, and the geodesic distance has restricted triangular inequality. Therefore, we do not consider this topology in this study.

## 6 CONCLUSION

**Summary:** In this work, we discuss the important properties of the feature embedding space for contrastive alignment. We show that the most commonly adopted cosine similarity has disadvantages in dealing with noisy data and training stability. Therefore, we propose to combine the oblique manifold with the negative inner product distance to tackle these problems. The proposed approach demonstrates better performance in the image-text retrieval tasks practically.

**Limiation:** First, due to remarkably limited computational resources, we cannot conduct experiments on a larger scale in terms of batch size, training data, and parameters in the neural network (and time). Second, in recent studies, besides the contrastive alignment, more pre-training tasks are appended to the head of the model using the non-normalized full token embedding. Such as image-text matching (Li et al., 2021a; Yang et al., 2022), image captioning (Yu et al., 2022), or masked modeling that do not employ the contrastive alignment (Wang et al., 2022). In these configurations, the performance improvement resulting from a better contrastive alignment could be marginal. And hence leave future work on designing the topology of the full token embedding.

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

# A Appendix

## A.1 Train hyper-parameters:

We train with a batch size of 2,048 and the AdamW optimizer in all our experiments. Following the hyper-parameters that improve training stability in the original CLIP (Radford et al., 2021) paper, we set the $\beta_1 = 0.9$, $\beta_2 = 0.98$, $\epsilon = 1e$-8, and weight decay $= 0.5$. We also employ the `gradient clip` operation with a global norm of 1.0. We did not employ mixed precision to reduce the possible variance introduced by random initialization. We apply a compact training scheme that updates the model for 108,000 iterations, which is roughly equal to training the model for 15 epochs of the dataset, and takes approximately 1 day to train one model using 32 Nvidia V100-16G GPUs. We use the cosine learning rate decay scheme of peak learning rate equal to $5e$-4, combined with a warmup period of 5,000 iterations. For data augmentation, we only apply the `RandomResizedCrop` with a scale range of $[0.5, 1.0]$.

In the linear probe evaluation, the hyperparameters follow the setup of MoCo v3 (Chen et al., 2021b). Concretely, we use SGD without momentum and no weight decay. The learning rate is schemed by cosine decay with a peak learning rate equal to 1.0, combined with a warmup period of 5 epochs. We train for 100 epochs and augment the image using the `RandomResizedCrop` with a scale range of $[0.75, 1.0]$ and `AutoAugment` with the code `rand-m9-mstd0.5-inc1`.

| Temp. Init. | Temp. Final | Converge Step | Zero-Shot I2T R@1 | Zero-Shot T2I R@1 | Zero-Shot Cls. Acc. | Linear Cls. Acc. |
|---|---|---|---|---|---|---|
| | | Topology: Sphere, | Distance: $-\boldsymbol{u}^T\boldsymbol{v}$ | | | |
| 2.659 | 4.033 | 18k | 48.3 | 31.45 | 30.62 | 60.38 |
| 5.310 | 4.021 | 39k | 46.8 | 31.13 | 29.60 | 59.82 |
| 1.000 | 3.976 | 22k | 49.0 | 30.33 | 28.59 | 59.56 |
| 1.000 | 1.000 | Detach | 5.1 | 3.461 | 4.04 | 45.37 |
| | | Topology: Euclidean, | Distance: $\|\boldsymbol{u} - \boldsymbol{v}\|_2$ | | | |
| 2.659 | 2.067 | 21k | 47.9 | 32.29 | 30.36 | 59.90 |
| 5.310 | 5.107 | 1k | 48.5 | 32.69 | 30.68 | 59.74 |
| 1.000 | 1.668 | 25k | 47.4 | 30.71 | 29.85 | 60.09 |
| 1.000 | 1.000 | Detach | 47.6 | 30.43 | 29.51 | 59.20 |
| | | Topology: Ob(64, 8), | Distance: $\mathrm{Geo}(\boldsymbol{u}, \boldsymbol{v})$ | | | |
| 2.659 | 3.135 | 20k | 50.7 | 32.35 | **30.79** | 60.43 |
| 5.310 | 3.168 | 55k | 50.7 | 31.59 | 30.34 | 59.60 |
| 1.000 | 3.024 | 42k | 49.9 | 32.49 | 30.21 | 60.61 |
| 1.000 | 1.000 | Detach | 4.1 | 2.921 | 3.10 | 21.67 |
| | | Topology: Ob(64, 8), | Distance: $-\mathrm{tr}(\boldsymbol{u}^T\boldsymbol{v})$ | | | |
| 2.659 | 2.231 | 24k | **52.3** | 32.89 | 30.70 | 60.32 |
| 5.310 | 2.280 | 57k | 50.3 | **33.37** | 30.23 | 59.76 |
| 1.000 | 2.174 | 36k | 50.9 | 32.71 | 30.50 | **60.66** |
| 1.000 | 1.000 | Detach | 30.3 | 18.48 | 21.20 | 57.93 |

Table 5: The retrieval and classification performance of different configurations under different temperature initialization conditions. The performance report in this table is the same as Table 2, but is aggregated by topologies. "Temp. Init." denotes the values for initializing temperature; "Temp. Final" denotes the final temperature at the end of training; "Converge Step" denotes the number of steps for tempearture starts to converge (changes less than 2% for an epoch.)

## A.2 Table 2 from the View of Topologies

In Table 5, we review Table 2 by the topologies. We further provide the final temperature at the end of training and at what step the temperature converges (changes less than 2% for an epoch, also see Figure 2). It can be seen that the performance of the Euclidean topology is only slightly affected by the initialization of the temperatures, and even though the temperature is detached from learning, it still performs reasonably well because of the unlimited distance range. While the spherical and oblique topologies are affected by how the temperature is initialized. However, a rough trend can

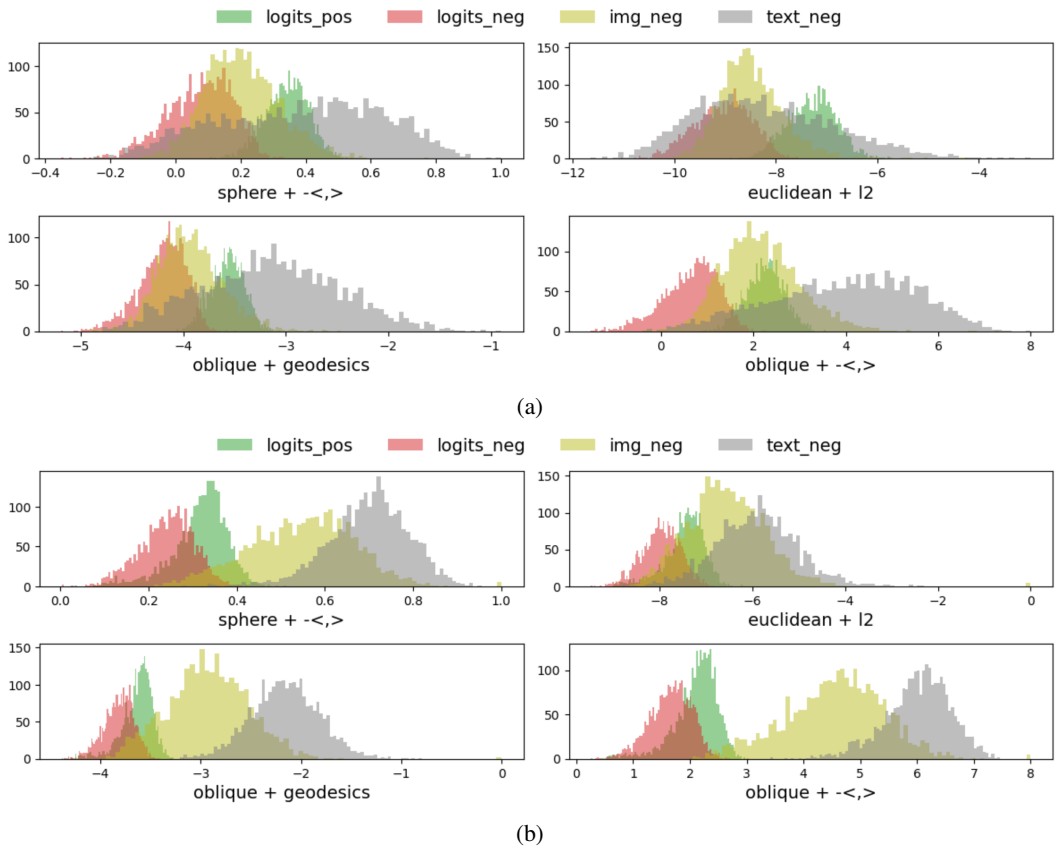

Figure 3: Visualization of the distribution of distances between samples. The `logits_pos` and `logits_neg` denote the distances between positive and negative image-text pairs, respectively. The `img_neg` and `text_neg` denote the distances between negative image-image and text-text pairs, respectively. The models are trained using the Yfcc datasets, (a) and (b) depict the distribution of in-domain data (Yfcc) and out-domain data (RedCaps), respectively.

be seen that the faster the temperature converges, the better performance the model achieves, which means the learnable temperature delays the learning of the methods. The model needs first to find a proper temperature and then begin to learn representations well.

## A.3 DISTRIBUTION OF LEARNED DISTANCE

We depict the distribution of distance for pairs of samples in Figure 3. As argued in Section 3.2 of the main manuscripts, since the cross-modal contrastive loss does not handle the uni-modal data distributions, the distance between negative pairs of images and texts could be much smaller than that of a positive image-text pair, resulting in a tighter distance bound. Also, we can see this phenomenon is much more severe in out-domain data, which could reduce the transferability of the feature embeddings to downstream tasks. It is also notable that, the oblique endowed with the negative inner product as the distance function learns similar distributions compared to the sphere reference, while the numerical values of distances between samples are inherently larger without having multiplied with temperature.

## A.4 ADDITIONAL ABLATION ON OBLIQUE STRUCTURE

We provide more ablation results regarding the structure of the oblique manifold under fixed total dimensions in Table 6. We can observe that the Ob(32, 32) configuration performs the best in general, while the sphere with more 1024-dimensional embedding has slightly better linear probe perfor-

| Topology | Distance | Zero-Shot I2T R@1 | Zero-Shot T2I R@1 | Zero-Shot Cls. Acc. | Linear Probe Cls. Acc. |
|---|---|---|---|---|---|
| Temperature, init=$e^1$, gradient=True | | | | | |
| Sphere(512) | $-\boldsymbol{u}^T\boldsymbol{v}$ | 48.3 | 31.45 | 30.62 | 60.38 |
| Sphere(1024) | $-\boldsymbol{u}^T\boldsymbol{v}$ | 50.7 | 32.05 | 29.60 | **60.53** |
| Ob(128, 8) | $-\text{tr}(\boldsymbol{u}^T\boldsymbol{v})$ | 49.4 | 32.85 | 30.55 | 60.12 |
| Ob(64, 16) | $-\text{tr}(\boldsymbol{u}^T\boldsymbol{v})$ | 50.3 | 33.25 | 30.34 | 60.16 |
| Ob(32, 32) | $-\text{tr}(\boldsymbol{u}^T\boldsymbol{v})$ | **52.3** | **33.47** | **30.62** | 60.32 |

Table 6: The retrieval and classification performance of the proposed approach using different oblique manifold structures and the multi-token implementation. "gradient={True/False}" donates if the temperature is learnable.

mance. We also notice that a more complicated structure provides better text-to-image retrieval results.

## A.5 DETAILED MODEL HYPERPARAMETERS OF OUR CLIP MODEL

| Hyperparameters | Value for Naive CLIP | Value for CLIP-Multi(16,32) |
|---|---|---|
| Batch size | 32768 | 32768 |
| Vocabulary size | 30522 | 30522 |
| Training epochs | 32 | 32 |
| Number [CLS] Tokens | 1 | 16 |
| Projection dims | 512 | 32 |
| Maximum temperature | 100.0 | 3.95 |
| Weight decay | 0.2 | 0.2 |
| Warm-up iterations | 2000 | 2000 |
| Adam $\beta_1$ | 0.9 | 0.9 |
| Adam $\beta_2$ | 0.998 | 0.998 |
| Adam $\epsilon$ | $10^{-8}$ | $10^{-8}$ |

Table 7: Hyperparameters used for training.

## A.6 ADDITIONAL RESULTS

We combine our proposed method with the TCL model Yang et al. (2022), which is one of the state-of-the-art vision-language retrieval models that employ contrastive visual-textual alignment in its earlier stage. During the pre-training, the TCL induces a mixture of in-modal and cross-modal contrastive losses, while conducting the masked language modeling (MLM) and image-text matching tasks simultaneously. During the testing, the cross-modal contrastive alignment head first lists sample pairs with high similarity scores, then these pairs are fed into the matching head to obtain the final matching scores. We alternate the topologies of all the embedding spaces with Ob(128,2); more precisely, we change the normalization function as shown in Section 3.1. For the experimental analysis in this subsection, we follow the configurations of the reference models, employ a collection of CC3M (Sharma et al., 2018), Coco captions (Chen et al., 2015), Visual genome (Krishna et al., 2017) and SBU (Ordonez et al., 2011) as the pre-training dataset, which contains roughly 4 million annotated image-text pairs. The models are then evaluated using Flickr30k (Plummer et al., 2015) and Coco captions (Chen et al., 2015).

The results are shown in Table 8. Since our method does not affect the matching head, we also report the performance of the contrastive alignment head. In general, our method improves the average recall performance, but the improvement is not significant. We consider the reasons as i) The method (or recent similar methods) employs pre-trained vision and language models, as well as a matching head and an MLM head, hence it is less sensitive to the gradients from the contrastive alignment; ii) The datasets employed for training contain less noise, while the training is scheduled with an overlength scheme (the zero-shot performance does not increase in the last 5 epochs).

| Method
*baseline[impl.]* | Flickr
I2T
R@1 | T2I
R@1 | Recall
mean | Coco
I2T
R@1 | T2I
R@1 | Recall
mean |
|---|---|---|---|---|---|---|
| *Zero-shot performance.* | | | | | | |
| TCL[official] | 93.00 | 79.60 | 93.97 | 71.40 | 53.50 | 79.49 |
| | (84.20) | (67.10) | (88.45) | (55.40) | (40.80) | (69.92) |
| TCL[our-impl.] | 91.00 | 78.28 | 93.25 | 70.16 | 53.05 | 79.07 |
| | (83.30) | (68.40) | (88.73) | (57.34) | (43.21) | (71.31) |
| TCL[Ob(128,2)] | 91.20 | 78.14 | **93.29** | 70.14 | 53.35 | **79.14** |
| | (84.80) | (67.86) | (88.84) | (57.10) | (43.13) | (71.32) |
| *Fine-tuned performance.* | | | | | | |
| TCL[official] | 94.90 | 84.00 | 95.57 | 75.60 | 59.00 | 82.87 |
| | (87.90) | (71.38) | (90.92) | (65.34) | (48.94) | (76.53) |
| TCL[our-impl.] | 93.80 | 83.06 | 95.17 | 73.56 | 57.74 | 82.06 |
| | (88.30) | (72.94) | (91.27) | (66.98) | (50.34) | (77.43) |
| TCL[Ob(128,2)] | 93.80 | 82.90 | **95.18** | 74.78 | 57.72 | **82.13** |
| | (88.60) | (73.26) | (91.39) | (65.60) | (49.83) | (76.86) |

Table 8: Retrieval performance on Flickr and Coco of our implemented TCL model and the variant using our proposed method. The numbers in brackets are the performance obtained using the contrastive alignment head.

**Additional Notes on TCL** We also provide the comparison results with officially released checkpoints. It can be seen that our implementation performs 0.5-1.0% worse than the official checkpoints. On the other hand, our implementation has better alignment head performance. Since we are employing the codes released in the official repository, the reason might be the following: i) Datasets difference, that we have ∼3000 fewer images in the SBU dataset while owning 5000 more images in the CC3M dataset; ii) We resize the CC3M dataset to short edge 500 pixels, while the official repository does not clearly provide the pre-processing approach; iii) We implicitly have a short training time or smaller matching loss weight than the official checkpoints due to the difference in the framework.

