# OpenReview forum: "Design of the topology for contrastive visual-textual alignment"
_ICLR.cc/2023/Conference — Submitted to ICLR 2023_

### Official Review · Reviewer_YGrh · 2022-10-21

**Confidence:** 4
**Correctness:** 4
**Technical Novelty And Significance:** 3
**Empirical Novelty And Significance:** 4
**Recommendation:** 3

**Clarity, Quality, Novelty And Reproducibility:**

This paper is badly written and hard to follow.  The idea is straightforward and lacks novelty. The authors impose the proposed method related to manifold learning, but it is not well proven and discussed theoretically.

**Strength And Weaknesses:**

Strength:
This paper proposed a simple strategy to make the temperature learnable. The proposed method simply changes two lines of the training codes to improve the performance.

Weakness:
1. This paper is badly written and hard to follow.
2. The key idea of the paper is to make the temperature learnable, which is straight and lacks novelty.
3. The proposed method is pretended too much. The authors impose the proposed method related to manifold learning, but it is not well proven and discussed theoretically.
4. The experiments cannot comprehensively demonstrate the effectiveness of the proposed method. From the experimental results, one could see that the proposed method could only improve slightly.


**Summary Of The Paper:**

This paper proposed a strategy to make the temperature learnable.

**Summary Of The Review:**

This method is too simple and lacks novelty. The authors impose the proposed method related to manifold learning, but it is not well proven and discussed theoretically. The experiments cannot well demonstrate the effectiveness of the proposed method.

---

> ### Author Response · Authors · 2022-11-18
> **Thank you.**
>
> Thank you for your comments, we wish you have a nice day.

---

### Official Review · Reviewer_W5p4 · 2022-10-24

**Confidence:** 5
**Correctness:** 2
**Technical Novelty And Significance:** 2
**Empirical Novelty And Significance:** Not applicable
**Recommendation:** 5

**Clarity, Quality, Novelty And Reproducibility:**

Quality: The paper is likely to have a modest impact on the community. Clarity: The paper is well organized but the presentation has minor details that could be improved. Originality: The main ideas of the paper are not novel or have limited novelty. Please see the weaknesses.

**Strength And Weaknesses:**

##Strength：
The proposed new distance metric is neat and could be easily used for the contrastive learning paradigm.

##Weaknesses:
Although the proposed method could be easily generalized to the existing multi-modal pretraining works using the contrastive learning paradigm, the effects cannot be verified well in the provided experiment results. As shown in Table 4, using the proposed metric only brings limited retrieval performance improvement compared to the vanilla baseline. The authors could provide more results to verify and defend the superiority of the proposed method.

Although the technical implementation of the proposed method is clear, there are some statements and arguments needed to be further claimed.
- “Here, the problem is that the numerical values of distances at equilibrium might be out of the distance range (e.g. [-1, 1] for the cosine similarity).” Why the numerical values of distances could be out of the range of [-1, 1] for the cosine similarity?
- “For instance, if there is a reasonable amount of false negative samples, then the model would learn a smaller negative distance for not being punished too hard when encountering false negative samples.”

- The reviewer finds that the font of the paper is different from other submission manuscripts. Does the paper use the proper template?


**Summary Of The Paper:**

This paper designs a new distance metric called Oblique(d/m, m) instead of the fashionable cosine similarity for the contrastive learning paradigm. To verify the effectiveness of the proposed simple method, the authors perform extensive experiments.

**Summary Of The Review:**

The paper is not well written and hard to follow. The experiments are not solid for comprehensively evaluating the proposed method.

---

> ### Author Response · Authors · 2022-11-18
> **Response to review W5p4**
>
> We sincerely appreciate the reviewer's very constructive comments and suggestions. The following is our response in order of questions and comments raised.
>
> ----
>
> > The authors could provide more results to verify and defend the superiority of the proposed method.
>
> We have added a new experiment with larger-scale datasets and models in Section 4.3. **Our ViT-B/16-Multi(32,16) model significantly outperforms models using the same visual backbone**, while even achieving remarkably better results compared to the official CLIP ViT-L/14 model, on zero-shot ImageNet, Flickr30K and Coco. Please check the revised paper for details. Although the study is not aiming for sota performance, we hope this experiment is sufficient in verifying the effectiveness of our proposed method.
>
> ----
>
> > “Here, the problem is that the numerical values of distances at equilibrium might be out of the distance range (e.g. [-1, 1] for the cosine similarity).” Why the numerical values of distances could be out of the range of [-1, 1] for the cosine similarity?
>
> ----
>
> Sorry for the confusion. We have updated this part to make it more clear. In contrastive learning, **since we employ the softmax and cross-entropy loss, the required numerical value of equilibrium is scaled by softmax, not the original distance**. That is, the equilibrium is achieved under the $\frac{\text{exp(positive value)}}{\text{sum of exp(negative value)}}$ term. The positive value and negative value might be out of the range of a distance function. For instance, the equilibrium needs a positive value of 10, and a negative value of 5, such that the noisy samples can be well-balanced during training. While the cosine similarity is constrained in [-1,1], so we need to scale the range by a learnable temperature to $[ -\tau, \tau]$.
>
> ----
>
> > “For instance, if there is a reasonable amount of false negative samples, then the model would learn a smaller negative distance for not being punished too hard when encountering false negative samples.”
>
> Sorry for the confusion. We have updated this part to make it more clear. In contrastive loss, we are simultaneously optimizing two targets, i) to make the positive pairs of samples closer; ii) to make the negative pairs of samples farther away from each other. If, in a dataset, we have false negative samples that are actually positive pairs. The loss target ii) will be very large. Therefore, the model will learn a smaller **positive** distance to lower the loss.
>
> ----
>
> > The reviewer finds that the font of the paper is different from other submission manuscripts. Does the paper use the proper template?
>
> Thank you very much for mentioning us. We tried the pdfLaTeX to compile the paper, it should look fine now.
>
> ----
>
> We are willing to address your further concerns on this paper.

---

> ### Author Response · Authors · 2022-11-23
> **Gentle Reminder**
>
> Dear reviewer W5p4,
>
> We sincerely appreciate the precious review time and valuable comments. Since we have provided corresponding responses, we hope to further discuss with you whether or not your concerns have been addressed. Please let us know if you still have any unclear parts of our work.
>
> Best Regards.

---

### Official Review · Reviewer_bLt8 · 2022-10-25

**Confidence:** 4
**Correctness:** 3
**Technical Novelty And Significance:** 3
**Empirical Novelty And Significance:** 3
**Recommendation:** 6

**Clarity, Quality, Novelty And Reproducibility:**


The paper is clear written and easy to follow. The problem and methodology are nicely formulated.
If I have to name a few flaws in presentation, there are some symbols do not can clear explanations. For example:
- Eq. (2)  H(.|.) denotes cross-entropy loss.
- [CLS] in section 3.1.

The idea sound novel, but the motivation needs to be strengthened.
The reviewer trust the reproducibility of this work.

**Strength And Weaknesses:**

+ In general, this paper is easy to read. Formulation of the problem and description of the method are clear and rigorous.
+ The idea of using oblique manifold to learn a visual-textual embedding space is novel.
+ The finding on learnable softmax temperature is a distance scaling factor of contrastive learning on noisy dataset is interesting.

- Missing some related literatures such as hyperbolic embedding space.

The motivation in choosing oblique manifold is a bit weak to some extent. The main arguments are 1) its geometry is spherical, where proper definition of uniformity is available, and 2) inner product saves computational cost.
- Why the uniformity in embedding space is a valid goal to embed text, given the structure of text can be hierarchical?
- While it is true that unbounded Euclidean space does not have proper uniform distribution defined (Sec 3.2 ii), no neural network is able to learn in unbounded space. We can simply clip to bounded range where uniform distribution is available.
- In terms of computational cost, do authors take into consideration l2 normalization steps in their algorithm?
- Also, is the computation of distance function a bottleneck for optimizing underlying deep learning models? It sounds like a these difference in computational cost are marginal.
- The statement of "numerical values of distances at equilibrium might be out of the distance range (e.g. [-1,1] for the cosine similarity)" is somehow unclear for me. Any two vectors naturally have a cosine similarity value constrained in [-1,1]. Why it might be out of range?




**Summary Of The Paper:**

This work strives to learn a visual-textual embedding space from weakly label image-text pairs from the internet.
In a typical contrastive learning with cosine-similarity, the noise in training data prevents the deep network to produce a solution with alignment-uniformity, and the network achieves a suboptimal system of equilibrium instead. This work resorts to oblique manifold as embedding space that tackles the equilibrium problem. Empirical improvement on zero-shot image to text retrieval task is observed.

**Summary Of The Review:**

This paper proposes to learn visual-textual embedding space on oblique manifold. The idea is interesting and novel, and the paper is well written. The reviewer feels pleasant to read the paper. Waiting to see the clarification of motivations and some statements.

---

> ### Author Response · Authors · 2022-11-18
> **Response to reviewer bLt8**
>
> We sincerely appreciate the reviewer's very constructive comments and suggestions. The following is our response in order of questions and comments raised.
>
> ----
>
> > Missing some related literatures such as hyperbolic embedding space.
>
> Thank you very much. We have updated the related work section. The hyperbolic embedding space has similar disadvantages to Euclidean space, therefore we do not implement it.
>
> ----
>
> > The motivation in choosing oblique manifold is a bit weak to some extent. The main arguments are 1) its geometry is spherical, where proper definition of uniformity is available, and 2) inner product saves computational cost.
>
> We have made our motivations clear in the revised Section 3.2 and Table 1. That is, **the border distance range is the key** to handling the noisy dataset problem. While we still need the topology to have **favored properties (low computational cost, properly defined uniformity, relaxed inequality)** for contrastive alignment learning. Although a configuration with unfavored properties may also function, the performance will be degraded.
>
> ----
>
> > Why the uniformity in embedding space is a valid goal to embed text, given the structure of text can be hierarchical?
>
> This is a very crucial question. **The uniformity in embedding space is the necessary property for optimizing the contrastive loss**, but not for better feature representation. In this paper, we majorly tackle the noisy dataset problem of contrastive loss and its optimization progress. Therefore, how to employ the hyperbolic space to represent the hierarchical information better is another topic, and we leave this as future work.
>
> ----
>
> > While it is true that unbounded Euclidean space does not have proper uniform distribution defined (Sec 3.2 ii), no neural network is able to learn in unbounded space. We can simply clip to bounded range where uniform distribution is available.
>
> As discussed in Sec 3.2 ii, we can clip to bounded ranges for Euclidean space, such that we can distribute the samples uniformly in, for instance, hypercubic space. It needs additional tricks for the contrastive loss to achieve uniformity in this scenario. For instance, via optimal transport algorithm, however, it is computationally burdensome, therefore, we did not consider that in our study. Please check Chen et al. (2021a) for more details.
>
> ----
>
> > In terms of computational cost, do authors take into consideration l2 normalization steps in their algorithm?
>
> Thank you for mentioning us. The l2 normalization is considered to have a computational complexity of $2d$ flops, we need $2bd$ more flops in the sphere and oblique. Also, the l2 normalization only takes additional memory resource linear scaling to the batch size and the oblique $m$. But given the fact that the term $bm$ ($\sim 2^{20}$) is much smaller than the squared batch size $b^2$ ($\sim2^{30}$), it should be safe to ignore it.
>
> ----
>
> > Also, is the computation of distance function a bottleneck for optimizing underlying deep learning models? It sounds like a these difference in computational cost are marginal.
>
> As aforementioned, the consumed memory that scales with the squared batch size $b^2$ ($\sim2^{30}$) is a huge number. For instance, in practice, we calculated the similarity matrix of size local $\times$ global batch, which consumes 67MB of memory for the sphere. However, the Euclidean space will consume 34GB of memory since it scales with $b^2d$, which is not feasible for learning on GPU devices.
>
> ----
>
> > The statement of "numerical values of distances at equilibrium might be out of the distance range (e.g. [-1,1] for the cosine similarity)" is somehow unclear for me. Any two vectors naturally have a cosine similarity value constrained in [-1,1]. Why it might be out of range?
>
> Sorry for the confusion. We have updated this part to make it more clear. In contrastive learning, **since we employ the softmax and cross-entropy loss, the required numerical value of equilibrium is scaled by softmax, not the original distance**. That is, the equilibrium is achieved under the $\frac{\text{exp(positive value)}}{\text{sum of exp(negative value)}}$ term. The positive value and negative value might be out of the range of a distance function. For instance, the equilibrium needs a positive value of 10, and a negative value of 5, such that the noisy samples can be well-balanced during training. While the cosine similarity is constrained in [-1,1], so we need to scale the range by a learnable temperature to $[ -\tau, \tau]$.
>
> ----
>
> We are willing to address your further concerns on this paper.

---

> ### Author Response · Authors · 2022-11-23
> **Gentle Reminder**
>
> Dear reviewer bLt8,
>
> We sincerely appreciate the precious review time and valuable comments. Since we have provided corresponding responses, we hope to further discuss with you whether or not your concerns have been addressed. Please let us know if you still have any unclear parts of our work.
>
> Best Regards.

---

### Official Review · Reviewer_NGQV · 2022-10-25

**Confidence:** 3
**Correctness:** 3
**Technical Novelty And Significance:** 2
**Empirical Novelty And Significance:** 2
**Recommendation:** 5

**Clarity, Quality, Novelty And Reproducibility:**

This Paper is clearly written with good quality. The originality might be minor with using the Oblique Manifold in alignment learning. From the provided pseudo-code, this paper should be easy to reproduce.

**Strength And Weaknesses:**

Strengths:
- The proposed algorithm is easy to implement and proved effective with multiple datasets.
- This paper provides detailed discussion on the properties of the embedding topology and make fair comparison with other topologies including Sphere, Euclidean, and Oblique.

Weaknesses:
- In Introduction, the authors discussed the noise problem in contrastive alignment learning, and it seems that the proposed embedding space would handle the problem better than others. However, it may lack the relevant discussions in the body part or experiments.
- This paper claims that “the sphere and oblique manifold have a proper uniform distribution defined as the surface area measure”, which might be the reason why Sphere and Oblique performs better than Euclidean. However in Table 2, it seems that the conclusion would be largely affected by the Temperature init.
- It may lack the discussion/visualization on the comparison of the learned distribution with Sphere and Oblique, which should give a more clear explanation on why Oblique performs better than Sphere.
- Oblique Manifold has been utilized in other feature learning tasks, such as [i] and more discussions should be added.
[i] Transductive Few-Shot Classification on the Oblique Manifold, ICCV 2021.



**Summary Of The Paper:**

This paper proposes to employ the oblique manifold as the embedding space for contrastive visual-textual alignment learning, where the feature representations are mapped onto the oblique manifold endowed with the negative inner product as the distance function. Discussions and experiments on multiple different topologies endowed with different distances demonstrates the superiority and effectiveness of the proposed approach.

**Summary Of The Review:**

Overall, this paper utilized a new distance function for contrastive visual-textual learning, although simple and effective, this paper does not provide new things up with Oblique Manifold, and it lacks deeper analysis/demonstration on how the method solves existing problems.

---

> ### Author Response · Authors · 2022-11-18
> **Response to reviewer NGQV**
>
> We sincerely appreciate the reviewer's very constructive comments and suggestions. The following is our response in order of questions and comments raised.
>
> ----
>
> > In Introduction, the authors discussed the noise problem in contrastive alignment learning, and it seems that the proposed embedding space would handle the problem better than others. However, it may lack the relevant discussions in the body part or experiments.
>
> We have made our motivations clear in the revised Section 3.2 and Table 1. That is, the border distance range is the key to handling the noisy dataset problem. While we still need the topology to have favored properties (low computational cost, properly defined uniformity, related inequality) for contrastive alignment learning.
>
> ----
>
> > This paper claims that “the sphere and oblique manifold have a proper uniform distribution defined as the surface area measure”, which might be the reason why Sphere and Oblique performs better than Euclidean. However in Table 2, it seems that the conclusion would be largely affected by the Temperature init.
>
> We have provided more discussion on this point in Section A.2. **In this paper, we do not claim "some topologies are strictly better than others". We emphasize that if the configuration is in favor of the contrastive loss, then it is more likely to perform better.** In the experiments, we observed that i) Euclidean is only slightly affected by the initialization of the temperatures, and still functions well even the temperature is fixed at 1.0, because of the unlimited distance range, but in general, it performs worse than the sphere and oblique; ii) While the spherical and oblique topologies are affected by how the temperature is initialized, the key factor is how much time it will take for the model to find the proper temperature for the datasets, as a result, a better initialization helps to learn good representations.
>
> ----
>
> > It may lack the discussion/visualization on the comparison of the learned distribution with Sphere and Oblique, which should give a more clear explanation on why Oblique performs better than Sphere.
>
> We have provided more discussion on this point in Section A.3. **The major reason for choosing the oblique with the inner product is because this configuration has a border distance range than the sphere, while its other properties are just in same as the sphere**, which is in favor of the contrastive learning, as discussed in the introduction and Section 3.2. It will be essentially a good outcome if the distribution of distances of samples for the oblique is similar to the sphere.
>
> ---
>
> > Oblique Manifold has been utilized in other feature learning tasks, such as [i] and more discussions should be added. [i] Transductive Few-Shot Classification on the Oblique Manifold, ICCV 2021.
>
> Thank you very much. We have updated the related work section. In our work, we focus on aligning cross-modal representations using contrastive loss.
>
> ---
>
> We are willing to address your further concerns on this paper.

---

> ### Author Response · Authors · 2022-11-23
> **Gentle Reminder**
>
> Dear reviewer NGQV,
>
> We sincerely appreciate the precious review time and valuable comments. Since we have provided corresponding responses, we hope to further discuss with you whether or not your concerns have been addressed. Please let us know if you still have any unclear parts of our work.
>
> Best Regards.

---

### Author Response · Authors · 2022-11-18
**Summary of Revisions**

We sincerely appreciate the constructive comments and suggestions from reviewers NGQV, bLt8, and W5p4.

Below, we summarize the major revisions made to this paper.

1. We add a new experiment with larger-scale datasets and models to address the insufficient experimental verification problem. That is, we collect a dataset that is similar to the official CLIP and open_clip, then we optimize a **ViT-B/16**-Multi(32,16) model and verify its performance on zero-shot ImageNet, Flickr30K and Coco. This model achieves remarkably better results on these datasets, even compared to the official CLIP **ViT-L/14** model. **We will make our code and learned weights publicly available after the decision of this paper is made.**

2. We clarify several unclear motivations mentioned by the reviewers.

3. We discuss more related works mentioned by the reviewers.

4. We provide another version of Table 2 in the appendix to explain the temperature initialization problem better.

5. The revised version is now compiled by pdfLaTex to correct the formatting.

Best regards,

Authors

---

### Decision · Program_Chairs · 2023-01-20

**Decision:**

Reject

**Justification For Why Not Higher Score:**

The authors did respond to some of the concerns, but the reviewers still felt that the paper is not ready for publication.  The authors also noted that one of the reviewers (the lowest-score reviewer) had a review that was unfair; however, even if we discount this review (which seems possibly reasonable), the paper still doesn't have enough support to accept.

**Justification For Why Not Lower Score:**

N/A

**Metareview: Summary, Strengths And Weaknesses:**

Thanks for your submission to ICLR.

For strengths, the reviewers mostly felt that the paper was easy to read.  Some reviewers appreciated the novelty of the method.

For weaknesses, some reviewers felt that there was weak motivation, that some additional validation was required, and some claims were unsubstantiated.  One reviewer felt that the paper was not well written.

Overall, even after discussion, three of the four reviewers recommended rejecting the paper, and thus it is not yet ready for publication.